## Overview Review

**Cite this article:** de Bakker DM, Perry CT and Webb AE (2025). Quantifying coral reef accretion in a changing world: approaches, challenges and emerging opportunities. *Cambridge Prisms: Coastal Futures*, **3**, e15, 1–17 https://doi.org/10.1017/cft.2025.10005

coral reef accretion; sea-level rise; fossil reefs; carbonate budget; photogrammetry

**Corresponding author:**
Didier M. de Bakker;
Emails: d.m.de-bakker@exeter.ac.uk;
didierdebakker@gmail.com

# Quantifying coral reef accretion in a changing world: approaches, challenges and emerging opportunities

Didier M. de Bakker ⬤, Chris T. Perry ⬤ and Alice E. Webb

Geography, Faculty of Environment, Science and Economy, University of Exeter, Exeter, UK

## Abstract

The long-term development of coral reef frameworks and the net vertical accretion of reefs fundamentally underpins the provisioning of most reef-related ecosystem services. One area of particular concern at present is how rates of reef accretion are changing under ecological decline and what the consequences of this may be for the capacity of reefs to keep pace with near-future sea-level rise (SLR). This may have major implications for the capacity of reefs to maintain their coastal protective functions and to support reef island stability. Both are issues relevant to understanding future tropical coastal risk. Long-term (millennial timescale) rates of reef accretion are relatively well constrained, including through past periods of sea-level fluctuations. However, widespread and persistent ecological degradation of coral communities has caused many reefs to diverge significantly from their past accretion trajectories. This renders historical analogues increasingly unreliable for projecting future accretion potential. Addressing this necessitates a reorientation towards considering reef accretion rates across shorter (ecological to geomorphic) timescales, i.e., over years to multi-decades. This is essential if we are to better constrain contemporary reef accretion rate and SLR interactions at timescales relevant to predicting emerging coastal risks and understanding future implications for reef-derived benefits. Here, we review existing approaches for quantifying vertical reef accretion rates of modern reefs. These methods span data recovered from fossil outcrops or core-derived records, the conversion of carbonate budget data, direct in situ measurements and emerging remote sensing and image-based techniques. The review explores the advantages and limitations of these different approaches and outlines options for developing an integrated framework to link past, present and future reef accretion potential.

## Impact Statement

Coral reefs are highly biodiverse and valuable ecosystems, providing essential ecological and socio-economic services, including coastal protection, fisheries, tourism revenue and the formation of reef islands that support human habitation. The continued provision of these services relies on the maintenance of a complex reef framework that is generated by sustained net carbonate production and accumulation. However, net reef accumulation potential is increasingly threatened by widespread ecological degradation and is projected to decline further under climate change. Quantifying and predicting the impacts of these changes on reef accretion rates is essential, particularly for evaluating reef capacity to keep pace with future sea-level rise. This is integral information for enhancing coastal risk modelling and guiding effective restoration efforts. Whilst much of our knowledge of reef accretion rates has arisen from fossil reef studies, this source of data is increasingly less reliable as modern-day ecologies diverge from those of the past. This necessitates a reassessment of how best to constrain contemporary accretion rates in a rapidly changing world. Here, we review the current state of play regarding methodologies for quantifying modern reef accretion rates. We review the strengths, limitations of different approaches and consider their applicability across different spatial and temporal scales. Inherent within this is an acknowledgment of areas in need of further development and testing, but the overall aim is to set the stage for the development of more robust, data-driven strategies to improve the modelling of reef accretion trajectories.

## Introduction

Sea level rise (SLR) poses a direct threat to many low-lying coastal regions by increasing the risk of wave-driven inundation and shoreline erosion (Pearson et al., 2017; Vitousek et al., 2017; Nicholls et al., 2021; Kench, 2024). Along many tropical reef-fronted coastlines, the natural breakwater structures that reefs provide are critical for mitigating this threat (Reguero et al., 2019, 2021; James et al., 2023). This wave protective function is primarily controlled by two key geo-

ecological aspects of reefs: (1) the capacity for sustained reef surface growth (hereafter *reef accretion*; see Box 1) which modulates above-reef water depths; and (2) the maintenance of a physically complex reef framework structure, which increases bottom friction (Young, 1989; Lowe et al., 2005; Monismith et al., 2013). Wave attenuation potential typically diminishes significantly as cross-reef water depths increase, and structural complexity is reduced (Storlazzi et al., 2011; Harris et al., 2018). With above-reef water depths projected to increase by at least half a metre due to sea-level rise alone by the end of the century, severe ecological and socio-economic consequences are anticipated (e.g. Ferrario et al., 2014; Quataert et al., 2015; Reynolds et al., 2015; Beck et al., 2018).

Sustained reef accretion will be critical to limiting these SLR-related threats, but is increasingly threatened as coral cover on reefs declines in response to an interplay of direct human-induced and climate change-related stressors (Perry et al., 2008; Perry and Alvarez-Filip, 2019; Woodhead et al., 2019). There are two key aspects to this ecological change that are relevant here. The first is the decline in coral abundance and especially of prominent reef-building species. This directly reduces the input of new carbonate material to build the reef (see Box 1) and diminish reef structural complexity (Alvarez-Filip et al., 2009; Alvarez-Filip et al., 2013). The second relates to shifts in the balance of carbonate-producing and carbonate-eroding processes (Glynn and Manzello, 2015; Perry and Harborne, 2016; Schönberg et al., 2017). As the latter becomes dominant, the reef accretion potential typically declines. For diverse socio-economic reasons, there is thus a pressing need for better informed projections of how coastal wave exposure will change in the near future under the interacting effects of impaired reef accretion potential and accelerating SLR (Storlazzi et al., 2017; James et al., 2023; Toth et al., 2023b; Webb et al., 2023).

Here, we consider the benefits and limitations of the different approaches available for addressing the challenge of quantifying the rates at which modern reefs can accrete (Figure 1). First, we consider what insights we can obtain from long-term (millennial to centennial timescale) assessments of reef development through past glacial–interglacial sea-level cycles, and from the last post-glacial SLR period. We then consider methodologies that have been more recently developed to assess reef accretion within contemporary reef environments. These include carbonate budget-based conversions and direct in situ measurements of surface elevation change. Finally, we consider emerging technology-based approaches, including structure-from-motion algorithms and those utilising remote aerial platforms. This synthesis serves as an evaluation of the trade-offs that exist between these various methods and aims to identify key focus areas for better constraining modern reef accretion and future trend trajectories. Improved assessments of reef accretion responses to ecological change will critically enhance projections of coastal vulnerability and will help establish baselines for informing focused reef restoration strategies.

## Applicability of fossil reef deposits for understanding reef accretion–sea-level interactions

One major focus of reef geoscience research has been on trying to understand and explain where and when coral reefs formed, how styles of reef development have varied geographically, and at what rates reefs have accreted under past sea-level change (e.g. Budde-meier and Hopley, 1988; Montaggioni and Braithwaite, 2009). Resultant insights have been invaluable for understanding reef formation through past glacial–interglacial cycles, and through

---

**Box 1.** Navigating the terminology

It is important to standardise and clearly define the various aspects of <u>**reef accretion**</u>; here, we adopt the following definitions based on previously established terminology.

<u>**CALCIFICATION**</u> is the basic process by which reef-building organisms convert calcium ($Ca^{2+}$) and carbonate ($CO_3^{2-}$) ions from supersaturated seawater into calcium carbonate ($CaCO_3$), thereby building their skeletons and forming the foundational material for the reef framework. Corals are referred to as *primary producers*, which accumulate as in situ colonies or reworked fragments. Conversely, other calcifying organisms, such as crustose coralline algae, articulated red and green algae, worms, bivalves and foraminifera, are typically *secondary producers,* which either contribute additional skeletal carbonate to the substrate or produce sediment that can help stabilise the reef framework.

<u>**FRAMEWORK PRODUCTION**</u> refers to the development and maintenance of the physical reef structure through the accumulation and lithification of coral skeletons and carbonates from secondary producers. Reef framework forms the 3D reef habitat that is shaped by a wide range of physical, chemical and biological processes (e.g. coral morphologies; bioerosion; sediment contributions, storm impacts). The geomorphic complexity of reef frameworks supports key reef functions and plays a crucial role in coastal protection through frictional effects on waves. *Framework stacking porosity* refers to the proportion of the reef volume that consists of spaces between the primary coral framework, which may be occupied by sediment, rubble, or remain as voids. *Framework density* is the bulk mass of carbonate material per unit volume.

<u>**REEF ACCRETION**</u> describes the vertical buildup of the reef framework, resulting from the net inputs of: (1) constructive processes, including calcification and lithification; (2) the re-incorporation or infilling of local and allochthonous material into the framework; (3) destructive processes (physical erosion and bioerosion); and (4) the displacement of $CaCO_3$ material (sediment and rubble). The speed at which net vertical accretion occurs can be an order of magnitude lower than the growth rate of an individual coral, but this rate defines whether a reef system can keep up, catch up or give up in response to rising sea levels.

---

the last post-glacial period (Kiessling, 2009; Woodroffe and Webster, 2014). Diverse modes of vertical reef accretion have subsequently been identified (Davies and Hopley, 1983; Kennedy and Woodroffe, 2002; Dullo, 2005; Montaggioni, 2005), and these have also shown that rapid reef accretion can occur where sufficient accommodation space exists both vertically (largely controlled by sea level) and laterally (controlled by slope angle and angle of repose of the reef front) (Neumann, 1985; Wood, 1999; Roff et al., 2015; Gischler et al., 2023). These insights have arisen from assessments of different types of reef deposits and the use of various analytical approaches, but broadly encompass: (1) assessments of exposed fossil outcrops; (2) data derived from rotary drilling of reef sequences; and (3) data from push cores through more recent fossil sequences.

Fossil outcrops probably provide the most comprehensive records for constraining past reef accretion patterns. Across such sequences, the arrangement of corals and associated deposited material can be clearly discerned, enabling the visual evaluation of reef-building processes and the selective collection of samples for dating (Figure 2a). These outcrops primarily occur in areas of tectonic uplift (e.g. Barbados and the Huon Peninsula Papua New Guinea) where former reef deposits have been subaerially exposed. These provide unique perspectives on reef growth trajectories during successive glacial–interglacial cycles (Chappell, 1974; Pandolfi, 1996). Similarly, excellent reef crest and shallow reef habitat deposits dating from the end of the last interglacial (MIS5e) occur in several regions, including the Bahamas, Curaçao, Mexico and Western Australia (e.g. Hearty and Neumann, 2001; Pandolfi and Jackson, 2001; Hearty et al., 2007; Blanchon et al., 2009; O'Leary

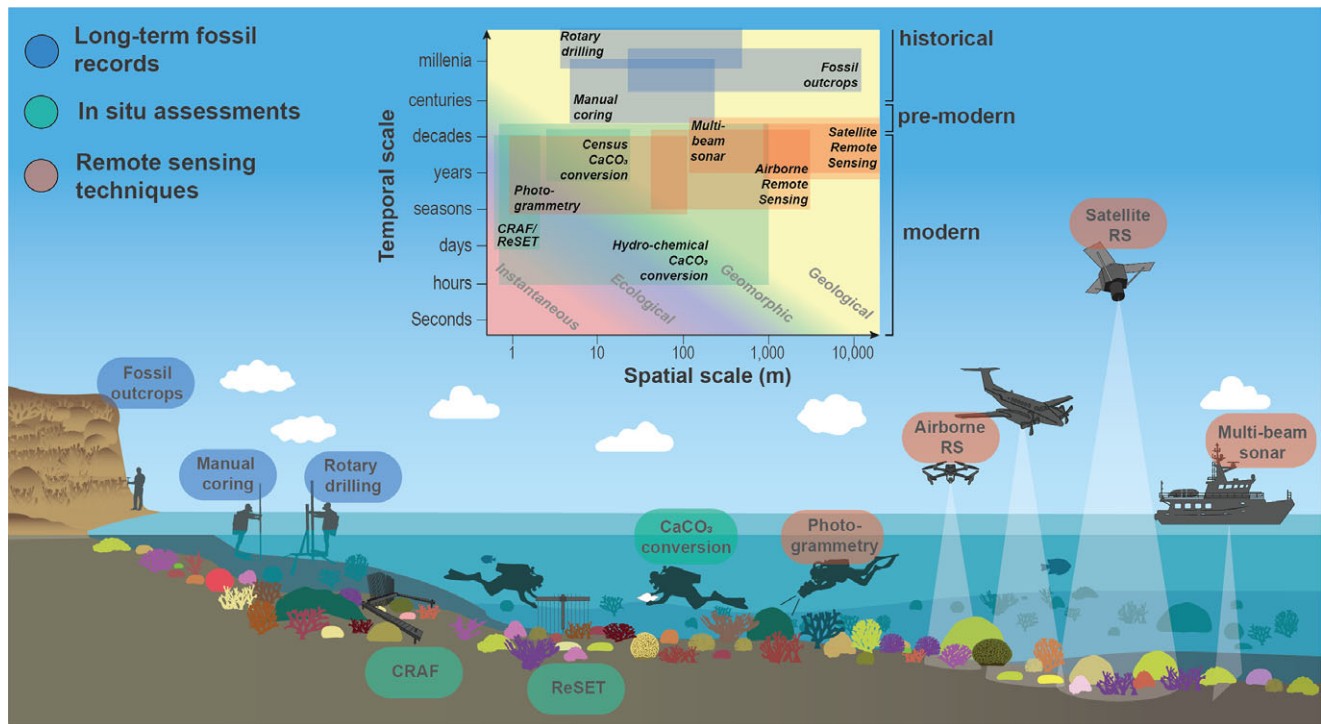

**Figure 1.** Overview of methods currently used to determine reef accretion rates in fossil reefs or to measure reef accretion in modern reefs. The inset, adapted from Perry et al. (2008), positions these methods within a framework of spatial and temporal scales, ranging from instantaneous (days/weeks) to geological (10,000+ years) processes, highlighting their operational contexts across reef environments.

et al., 2013; Muhs, 2022). Despite diagenetic issues and associated dating challenges, these fossil sequences have significantly advanced our understanding of reef behaviour under highstand sea-level oscillating conditions, and under sea-level states of several metres above present (Blanchon and Shaw, 1995; Blanchon and Jones, 1997; Hearty et al., 2007; Hubbard, 2009). As such, these are potential analogues for more extreme future SLR states. Of particular note are several mid- to late Holocene reef outcrops with exceptionally preserved coral assemblages that have relevance to modern compositions (e.g. in the Huon Peninsula, Dominican Republic and Cuba) (Mann et al., 1984; Taylor et al., 1985; Blanchon et al., 2009; Greer et al., 2009).

The expansive nature of many of these sequences has been of considerable value for understanding past spatial coral community configurations. Specifically, the shallow water deposits, which are often encountered in outcrops, have arguably been the most relevant for understanding reef accretion behaviour in a sea level change context. However, the bathymetrically restricted extent of most fossil outcrop sequences typically precludes assessment of long-term reef accretion rates. Deep rotary drill cores have played an important role in addressing this gap (Macintyre, 1975). These facilitate reconstruction of reef development from their initiation to the sub-present (Davies and Hopley, 1983; Montaggioni, 2005; Hynes et al., 2024), with the longest cores penetrating through to pre-Holocene strata. Outcrops may thus provide higher-resolution evidence of the structural and ecological configurations of past reefs, while deep reef cores extend our understanding across broader paleoenvironmental and palaeoecological timescales (Dullo, 2005).

With the capacity of spanning $10^3$- to $10^4$-year intervals (Figure 2), core-based geological reconstructions have substantially improved our understanding of reef-system scale responses to SLR (Woodroffe and Webster, 2014). The inferred responses of reef communities to

climatic change and varying local environmental conditions have provided the basis for establishing an increasingly robust baseline for modern reef accretion dynamics. However, these core-based records are not without challenges or limitations, one of which is the narrow diameter of cores (typically <10 cm) that limits spatial coverage (Hubbard et al., 1990). Additional issues arise with core recovery, dating accuracy, age reversals and the interpretation of diagenetic effects (Woodroffe and Webster, 2014). The inferred historical climate to reef growth relationship also reflects accretion patterns of pre-anthropogenic reef assemblages. These were generally considered to be resilient and largely capable of sustaining accretion rates that kept pace with or caught up with rapid sea-level rises (Buddemeier and Smith, 1988).

Over recent decades, percussion and push core approaches (Figure 1) have emerged as alternative tools for studying past reef accretion, particularly in sediment-rich deposits. These manually operated systems capture shorter intervals of reefs (up to ~6 m deep) but typically preserve 100% of the original sequence upon extraction, which is much higher than rotary drilling techniques (Aronson et al., 1998; Ryan et al., 2016; Perry et al., 2017). Faster deployment speeds can also enable higher sampling replication (sub-metre distances), with the possibility to develop more robust reconstructions of locality-specific framework accretion histories (Dardeau et al., 2000; Smithers and Larcombe, 2003; Perry et al., 2009). The value of this approach lies not only in the suitability for discerning reef development behaviour in sediment-rich, turbid or inshore reef environments, but notably in the potential to capture relatively good records from recent pre-modern periods (Partain and Hopley, 1989; Aronson et al., 2002; Perry and Smithers, 2006). This has been particularly useful for constraining reef responses to disturbances at geomorphic (100s–1,000s years) and ecological (up to 100 years) timescales

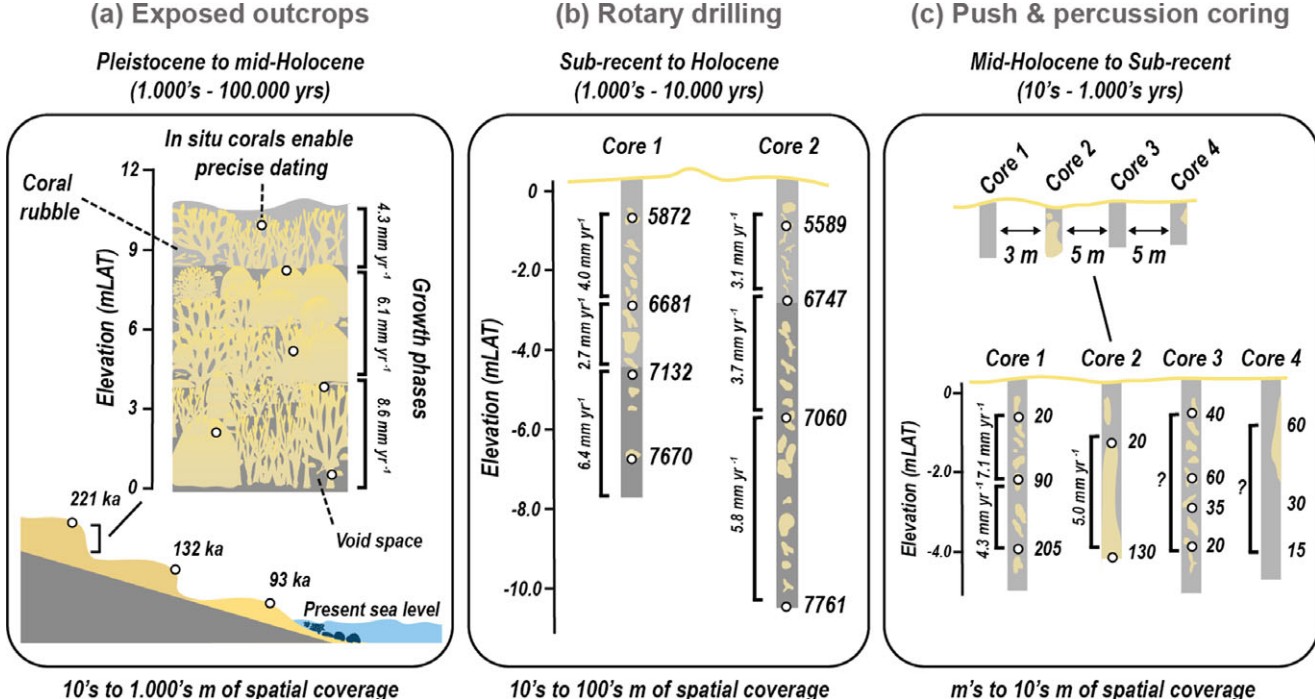

**Figure 2.** Summary of techniques utilising fossil reef deposits to ascertain historical rates of vertical reef accretion, visualised across scales of temporal coverage and timescales of interest. (a) Fossil reef outcrops, such as reef terraces formed through tectonic uplift, provide access to reef frameworks that developed at former sea-level highstands, often linked to glacial–interglacial cycles (e.g. Marine Isotope Stages). These outcrops allow the extraction of data on vertical reef accretion, paleo-ecological assemblages, framework porosity and the reconstruction of long-term reef development in response to sea-level changes. (b) Long rotary-drilled cores provide continuous vertical records of reef accretion over millennial to multi-millennial scales. These cores offer high-resolution age-depth profiles, enabling estimation of long-term average accretion rates. Dense sampling and deep penetration reduce the risk of age reversals and enhance interpretation of reef framework development, coral assemblages and internal structure. (c) Short percussion or push cores are used to assess accretion rates over shorter timescales (decades to centuries). Vertical age-depth profiles from these cores provide average accretion rates over recent intervals, though interpretation may be complicated by the nature of material recovered: *Core 2* captures a single massive coral colony preserved in situ; *Core 3* includes a storm-deposited clast assemblage, leading to unclear age progression; *Core 4* contains a mix of older in situ coral and younger rubble, complicating temporal interpretation. Elevation in each panel is shown relative to metres above Lowest Astronomical Tide (mLAT), and all ages are reported in millennia (ka).

(Figure 2), with some examples even including evidence of reef responses to early anthropogenic pressures. Studies in Belize, Panama and the inshore Great Barrier Reef, for instance, have linked periods of accretionary stasis, sediment stress and transitions in species assemblages to early human settlement (Cortés et al., 1994; Greenstein and Pandolfi, 1997; Perry et al., 2012b; Roff et al., 2013).

Despite advancements in both coring and radiometric dating, and the growing core record, accurate estimates of reef accretion rates from the most recent past (decades to centuries) remain rare (Perry et al., 2013b; Morgan et al., 2016). A degree of inference therefore remains inherent to the interpretation of fossil record deposits, and uncertainties in dating can easily be introduced through misinterpretation of factors such as age reversals, post-mortem diagenetic transformations, heterogeneous sedimentary facies and susceptibility to post-depositional processes (Scoffin, 1992; Edinger et al., 2007; Perry and Hepburn, 2008) (Figure 2). To further complicate the interpretation of recent facies, these issues are often most evident closer to the top of cores. Additionally, because of logistical constraints, information gaps persist for remote and offshore regions and harder to reach (e.g. high-energy reef crest or mesophotic) reef zones (Sherman et al., 2023).

Perhaps most critical, given the focus of this review, fossil-based reef accretion records derive from reefs, which often were accumulating under either different environmental and/or ecological conditions from those being experienced today. For example, the early post-glacial phase was characterised by periodic rapid "jumps" in

sea level, whilst more recent phases of reef growth (e.g. over the past ~6,000 years) have occurred under regionally divergent sea-level states and rates of change (Mann et al., 2019). These observations do not negate the value of past reef accretion records in a contemporary change context; at worst, they provide invaluable baseline reference points against which to assess modern accretion rates and styles. However, the speed of contemporary reef change implies we must acknowledge these differences when drawing assumptions from past accretion behaviours (Dougherty et al., 2019).

## Carbonate budgets and estimates of contemporary reef accretion potential

Given the rapid pace of recent ecological change and the recognition that palaeo- and historic reef behaviours may become increasingly less reliable analogues for present-day reef conditions, there is a growing incentive for quantifying reef accretion potential within the context of modern ecological settings. At present, this is typically addressed by considering the net accumulation of calcium carbonate by the active reef community. Net carbonate production rates per unit of reef area (G, in kg $CaCO_3$ m$^{-2}$ yr$^{-1}$) represent the sum of gross carbonate produced by corals and secondary calcifiers, less the carbonate lost through physical, chemical and biological erosional processes (Land, 1979). This "carbonate budget" is typically quantified either through a census-based or hydro-chemical approach (Perry et al., 2008; Browne et al., 2021). Resultant rates of

net carbonate production can then be converted into estimates of vertical reef accretion potential (in mm yr$^{-1}$) by considering the density of accumulated CaCO$_3$, the framework stacking porosity and the retention of framework-derived CaCO$_3$ sediments (e.g. Chave et al., 1972; Smith and Kinsey, 1976; Hubbard et al., 1990; Kinsey and Hopley, 1991; Perry et al., 2013a, 2018).

Pioneering efforts to quantify net reef accumulation in this context relied on reef-scale biological data to underpin carbonate budget estimates, with some of the earliest rates (in mm yr$^{-1}$) given by Chave et al. (1972 and references therein). Whilst these early efforts offered valuable perspectives on the links between gross and net carbonate production and potential net accumulation rates in modern reef settings, they typically relied on crude estimates of community-scale processes. These early budget calculations were, for example, generally skewed towards visibly dominant accretional processes; gross CaCO$_3$ production by standing stocks of corals and calcifying algae. Underpinning process data were often derived from different localities, and major geographical generalisations

were necessary. Advances to these approaches were therefore made using more process-based concepts. These included addressing data gaps related to secondary producers (Stearn et al., 1977; Scoffin et al., 1980), the (re-) integration of detrital material (sediments and rubble) into and export from the reef framework (Land, 1979; Hubbard et al., 1990), and refining estimates for local calcification and bioerosion rates (Scoffin et al., 1980; Sadd, 1984; Hutchings, 1986). Several studies also examined how local factors such as energy conditions and reef morphology might shape carbonate production across distinct reef habitats (e.g. Scoffin et al., 1980). Resultant budget estimates culminated in much better-constrained estimates of modern reef accretion. Nonetheless, these are still derived from often coarse cumulative approximations of gross accretional and erosional rates.

To circumvent the reliance on rate data for individual budget constituents, several early studies instead employed alkalinity reduction techniques to quantify net community-scale calcification (Smith, 1973; Kinsey, 1979; Kinsey, 1985a,b) (Figure 3a). This hydro-

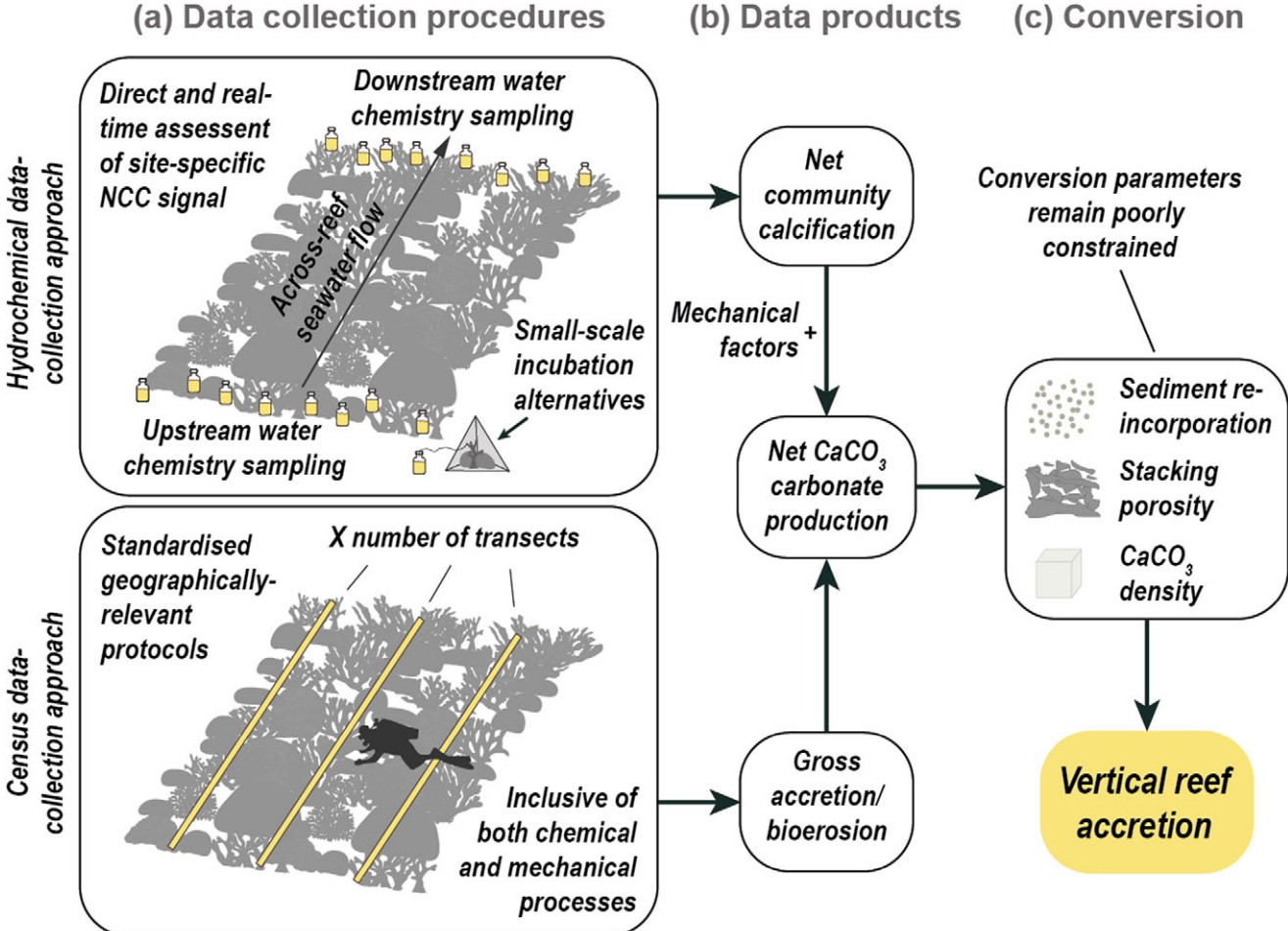

**Figure 3.** Graphical summary of the process of estimating vertical reef accretion rates (mm yr.$^{-1}$) from carbonate budget data. (a) Data collection methods include the hydro-chemical approach (top panel) and the census-based approach (bottom panel). The hydro-chemical method provides direct, real-time measurements of net community calcification (NCC), capturing all precipitation and dissolution processes at community scales. It is typically limited to reef areas with unidirectional flow or requires localised alternatives such as chamber-based or point-source sampling systems. The census-based approach uses standardised protocols applicable across regions and includes both biological and physical components of carbonate budgets. However, it is time-intensive, and parameterisation often relies on generalised values for certain taxa. Several other processes, such as passive dissolution or microbial lithification, are generally not incorporated into current protocols. (b) Data products from both methods differ in completeness and resolution. Hydro-chemical outputs are all-encompassing with regard to the chemical fraction, but do not resolve physical import/export. Census-based budgets can over- or under-estimate processes due to data gaps or a lack of temporal specificity. (c) Conversion to vertical accretion rates (mm yr$^{-1}$) requires integrating net CaCO$_3$ production with assumptions about sediment re-incorporation, framework porosity (ideally linked to coral community structure) and CaCO$_3$ density. This step is complicated by poorly constrained estimates for sediment dynamics (e.g. storm-driven removal or deposition), limited understanding of how different environmental conditions (e.g. reef assemblage or varying energy regimes) influence stacking porosity and variability in sediment retention.

chemical approach tracks salinity-normalised changes in seawater total alkalinity ($A_T$) driven by benthic calcification and erosion processes as water flows through the reef system. A decrease in $A_T$ indicates net positive calcification (i.e. $CaCO_3$ formation), and increases reflect net $CaCO_3$ dissolution. Resultant real-time $CaCO_3$ production derived from the reef community's chemical signal can subsequently be converted into estimates of vertical accretion (Smith and Kinsey, 1976; Kinsey and Hopley, 1991 and references therein). Critically, these studies were the first to consider community-specific framework stacking porosities in their calculations of vertical accretion rates (Figure 3c). These hydro-chemical-based measurements are particularly effective on reefs with linear flow conditions but are less reliable in environments where complex multi-directional currents and turbulent hydro-dynamics complicate modelling and interpretation (Courtney et al., 2016). In situ reef enclosure experiments (Figure 3a) may offer a more widely applicable alternative, yet at considerably reduced spatial scales (Van Heuven et al., 2018; Webb et al., 2021). More pertinent to reef accretion assessments, however, is that these approaches capture only the net effects of chemical processes, therefore not accounting for most mechanically driven processes (e.g. Courtney et al., 2016; Muehllehner et al., 2016). To address this, recent efforts have adopted multi-faceted methods that integrate census-derived factors to account for non-chemical variables (Courtney et al., 2022).

Both census- and hydro-chemical methodologies have thus provided useful insights into reef carbonate production rates and, by extrapolation, "modern" reef accretion rates. However, with growing recognition of the scale of ongoing reef ecological changes, the focus has shifted towards methods that more directly link reef accretion rates to specific ecological conditions. These approaches aim to capture ecological impacts on both carbonate-producing groups and eroding taxa. The early work of Eakin (1996, 2001), Edinger et al. (2000) and Bak (1976, 1994) was instrumental here and informed subsequent advancements made by Harney and Fletcher (2003), Hart and Kench (2007) and Mallela and Perry (2007). These culminated in the conceptual piece on the reef budget states under ecological change by Perry et al. (2008). Underpinning these approaches was a recognition of the need to record abundance data of all major accreting and bioeroding taxa and to apply, where possible, species-specific rates of calcification, linear extension and erosion. The *Reef-Budget* monitoring tool (Perry et al., 2012a), which arose from the above work, was designed to try and standardise census-based carbonate budget data collection. This monitoring system has since been applied at varying spatial scales (Perry et al., 2018; De Bakker et al., 2019; Molina-Hernández et al., 2020; Browne et al., 2021; Morris et al., 2022) and to discern the impacts of a range of disturbance events (e.g. Lange and Perry, 2019; Toth et al., 2023a; Lange et al., 2024).

Critically, in the context of this review, improving constraints on reef carbonate budgets (Lange and Perry, 2020; Browne et al., 2021) have also facilitated efforts to explore the consequences of reef ecological change for reef accretion rates (also described as maximum reef accretion potential, or $RAP_{max}$, sensu Perry et al., 2018). However, key aspects of the budget-to-accretion conversion remain reliant on often poorly constrained and non-locality-specific estimates of carbonate production or erosion processes. For example, recent efforts still rely on the initial – inadequately parameterised with regards to different ecologies – approximations of framework stacking porosity (Smith and Kinsey, 1976; Kinsey and Hopley, 1991). This can have a major bearing on the

final conversion calculations given the divergent ways that the skeletons of different coral taxa break down post-mortem (Davies and Hopley, 1983). Furthermore, a general paucity of empirical data limits our understanding of the complex processes of physical framework erosion (e.g. storm-driven) (Harmelin-Vivien, 1994; Perry et al., 2014), increasingly relevant chemical dissolution processes (Eyre et al., 2018; Dee et al., 2020; Doney et al., 2020), and the transport of (non)-bioerosional sediments and rubble (e.g. Hubbard, 1992; Perry, 1996; Blanchon et al., 1997; Harney and Fletcher, 2003; Kench and McLean, 2004; Morgan and Kench, 2014).

Only a limited number of studies have addressed regional modifications or incorporated local data to better constrain these aspects (e.g. Browne et al., 2013). Others have included specific parameters, such as refined $CaCO_3$ stacking porosity estimates, either tailored to local community compositions (Perry et al., 2018; Roff, 2020) or derived from regional core records (Toth et al., 2022, 2023b). Some studies have also factored for local environmental drivers, such as wave-energy regimes (Perry and Morgan, 2017) and terrigenous carbonate sediment inputs (Perry et al., 2012b; Januchowski-Hartley et al., 2020). However, considerate of its assumptions and limited temporal coverage (decadal scale), carbonate budget conversions have become established as the current best option for systematically quantifying modern reef accretion potential. Its application is further widely anticipated to be enhanced in both time and space by leveraging remotely sensed habitat mapping, high-resolution imaging techniques and automated tools for the analysis of image-based data (see following sections).

## Emerging techniques for direct measurement of vertical reef accretion

Issues inherent in budget-conversion approaches can theoretically be avoided by measuring actual reef surface elevation change in situ against a known fixed reference datum. To date, only a few studies have attempted to actively measure such in situ reef accretion but methodologies to address this are emerging and are being further developed to leverage data from remotely sensed platforms. As discussed below, challenges remain with both approaches, but both offer promising avenues for future research.

### In situ frame-based measure of accretion rates

Early inter-temporal measurements of reef surface elevation change were conducted by Eakin (1992, 1996), who measured reef flat height at predetermined points along a wire stretched between pairs of stainless-steel rods anchored in the reef framework. While this method provided actual measures for vertical substrate erosion by re-measuring the same points on the substrate over time, experimental longevity (~3 years) and accuracy were limited due to degradation of installed materials. The recently developed coral reef accretion frame (CRAF) offers a more robust alternative (Kench et al., 2022). This system enables repeatable millimetre-scale measurements of reef surface elevation using a relocatable measuring platform, with vertically adjustable measurement rods, mounted on four permanently fixed bolts (Table 1). Each deployment yields a permanent photo record of between 99 (a single line of measurement rods) and several hundred high-precision point measurements of the underlying substrate height across small (0.25 $m^2$) area plots. The CRAF was developed for deployment

**Table 1.** Summary of emerging approaches currently employed or with potential future application for measuring vertical reef accretion, arranged left to right by increasing spatial coverage and decreasing measurement resolution

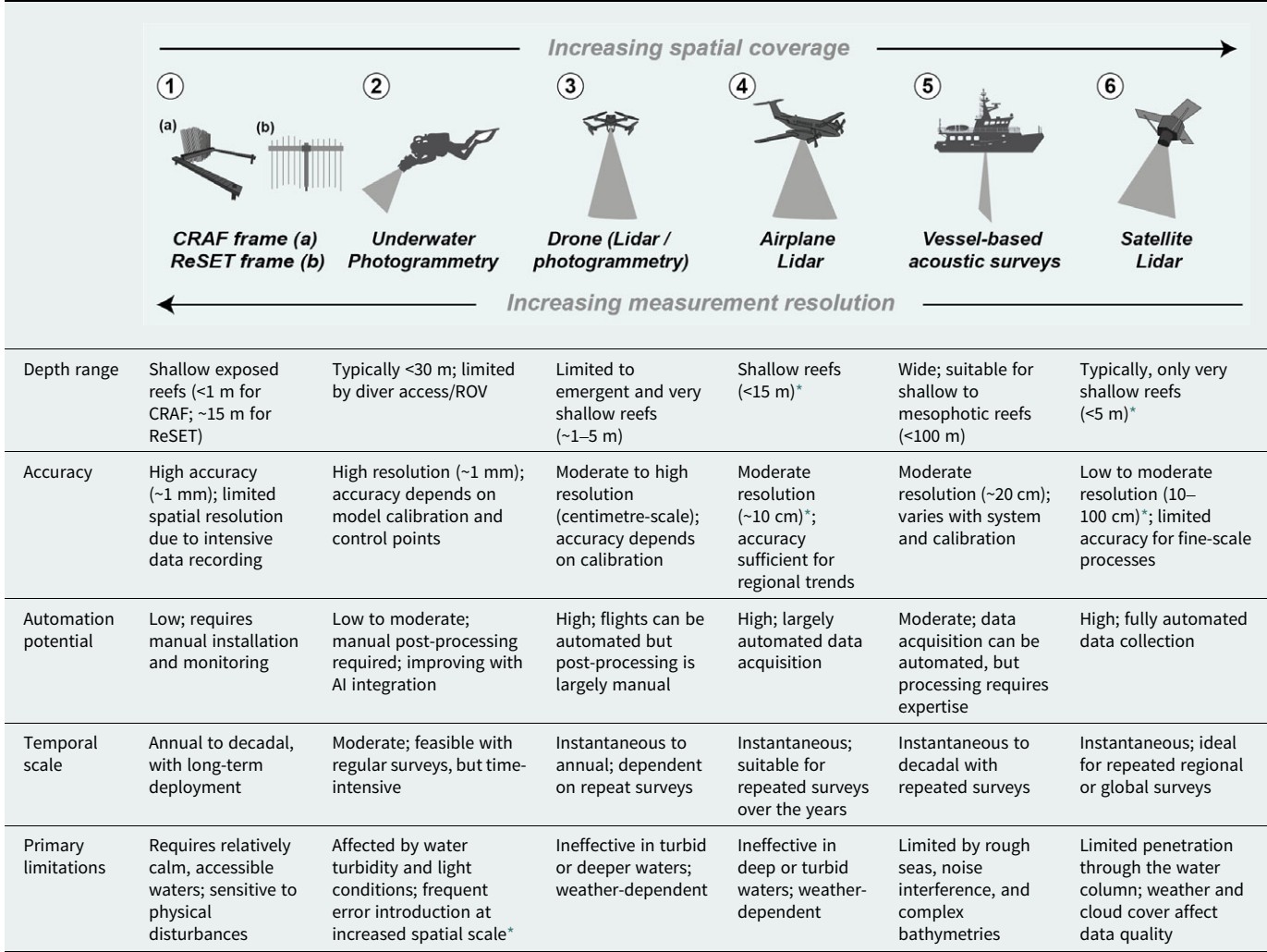

| | CRAF frame (a) ReSET frame (b) | Underwater Photogrammetry | Drone (Lidar / photogrammetry) | Airplane Lidar | Vessel-based acoustic surveys | Satellite Lidar |
|---|---|---|---|---|---|---|
| Depth range | Shallow exposed reefs (<1 m for CRAF; ~15 m for ReSET) | Typically <30 m; limited by diver access/ROV | Limited to emergent and very shallow reefs (~1–5 m) | Shallow reefs (<15 m)* | Wide; suitable for shallow to mesophotic reefs (<100 m) | Typically, only very shallow reefs (<5 m)* |
| Accuracy | High accuracy (~1 mm); limited spatial resolution due to intensive data recording | High resolution (~1 mm); accuracy depends on model calibration and control points | Moderate to high resolution (centimetre-scale); accuracy depends on calibration | Moderate resolution (~10 cm)*; accuracy sufficient for regional trends | Moderate resolution (~20 cm); varies with system and calibration | Low to moderate resolution (10–100 cm)*; limited accuracy for fine-scale processes |
| Automation potential | Low; requires manual installation and monitoring | Low to moderate; manual post-processing required; improving with AI integration | High; flights can be automated but post-processing is largely manual | High; largely automated data acquisition | Moderate; data acquisition can be automated, but processing requires expertise | High; fully automated data collection |
| Temporal scale | Annual to decadal, with long-term deployment | Moderate; feasible with regular surveys, but time-intensive | Instantaneous to annual; dependent on repeat surveys | Instantaneous; suitable for repeated surveys over the years | Instantaneous to decadal with repeated surveys | Instantaneous; ideal for repeated regional or global surveys |
| Primary limitations | Requires relatively calm, accessible waters; sensitive to physical disturbances | Affected by water turbidity and light conditions; frequent error introduction at increased spatial scale* | Ineffective in turbid or deeper waters; weather-dependent | Ineffective in deep or turbid waters; weather-dependent | Limited by rough seas, noise interference, and complex bathymetries | Limited penetration through the water column; weather and cloud cover affect data quality |

*Future resolution or range improvements likely as technologies develop.

in a shallow reef flat habitat, which allows data recording during low tide substrate exposure. While avoiding many of the obstacles of in-water data collection, this design is potentially less adequate for deployment in submerged reef habitats where even very modest water flow exerts considerable stress across measuring frames. The necessary 4-point fixture may also prove challenging in topographically complex locations or may need to be adapted for such conditions.

An alternative approach, with application within submerged locations, is the ReSET frame (Table 1). Based on the surface elevation table (SET) system, widely deployed in wetland and mangrove areas (e.g. Boumans and Day, 1993; Cahoon et al., 2002), it relies on a single pin permanently fixed in the reef substrate. This allows deployment in more structurally complex environments. Equivalent to the CRAF, the length of each of the 12 adjustable rods protruding from the top of a rotatable horizontal arm enables measures of the height of the underlying substrate. While yielding less measurements per unit area than the CRAF (48 in 0.5 m²), this method can be uniformly deployed in all hard-bottom inter- or sub-tidal environments. Data is collected visually in situ with sub-millimetre resolution. Capturing a permanent photographic record, akin to the CRAF

approach, is also possible, but only under exceptionally low-energy conditions.

These direct measurements of changes in reef elevation, inclusive of all relevant processes, produce accurate time series that facilitate comparisons of reef elevation across spatial gradients (e.g. ecological conditions) and temporal scales relevant to modern reef-building. While such direct measurements have the potential for broad application, the data collection process is generally time-intensive. Their value might therefore lie in targeted (geographical or specific reef component) studies rather than being routinely incorporated into standard field methods. One example might be to validate rate estimates derived using census approaches. Almost inherent to such manually operated systems, the error margins of both the CRAF and ReSET methods (~1–3 mm) are also roughly equal to the annual vertical change in reef elevation observed in many ecologically degraded reef habitats. Consequently, extended periods of data collection (>5 years) will probably be required to develop robust time-series datasets. However, once baseline measurements are established and if reference structures are properly maintained, these methods could provide valuable data spanning decadal timescales.

## Emerging technology-based approaches to quantifying reef accretion

Rapidly advancing imaging techniques may help address the spatial coverage constraints of both in situ and census-based studies (Table 1). Airborne or satellite remote sensing (multi- and hyperspectral imaging, LIDAR), Structure-from-Motion (SfM) photogrammetry, automated acoustic systems (multibeam sonar), or hybrids of these can generate robust uniform datasets with spatial coverage (10s to 10,000s of square metres) that significantly surpass traditional transect line- or quadrat-based survey techniques (Walker et al., 2008; Brown et al., 2011b; Goodman et al., 2013; Montes-Herrera et al., 2021; Teague et al., 2022; Remmers et al., 2024). With the potential for high temporal resolution, these remote sensing platforms can also more effectively capture impacts of stochastic events on vertical elevation change (e.g. hurricane-driven rubble deposition) (Blanchon et al., 2017). Critically, these technology-based approaches have the potential for cost-effective data acquisition and generate permanent and non-discriminative data records (Mumby et al., 1999, 2004b; Jupiter et al., 2013). This also permits the use of increasingly efficient and automated processing workflows (Hopkinson et al., 2020; Burns et al., 2022; Kopecky et al., 2023).

### Large spatial-scale remote sensing techniques to monitor reef elevation change

In coral reef research, remote sensing has predominantly been leveraged to advance detailed mapping of global reef systems, with particular relevance to remote or otherwise inaccessible areas (e.g. Purkis et al., 2019; Lyons et al., 2020; Kennedy et al., 2021; Nguyen et al., 2021; Barve et al., 2023). Remote sensing has also proved relevant in large-scale monitoring of specific ecological resilience indicators such as live coral cover, macroalgal abundance, habitat loss or the extent of bleaching events (Mumby et al., 2004a; Andréfouët et al., 2013; Knudby et al., 2014; Hedley et al., 2016; Parsons et al., 2018; Bakker et al., 2023). At present, however, these ex-situ techniques still exhibit diminishing resolution with expanding spatial coverage, depth and structural complexity (Goatley and Bellwood, 2011; Figueira et al., 2015). Further uncertainty is introduced by factors such as variable weather conditions, tides and non-uniformity in water column properties (Lesser and Mobley, 2007; Zawada et al., 2010; Hedley et al., 2016; Goodman et al., 2020). Accurately assessing vertical changes remains specifically challenging, as it requires high resolution data and relies on all bathymetric surveys to be referenced to a consistent tidal datum for meaningful comparison over time. These considerations might explain why remote sensing techniques have not, as yet, been broadly applied to quantify changes in reef surface height.

Yates et al. (2017) is, to our knowledge, the only study highlighting the potential of high-resolution reef bathymetry mapping to quantify reef elevation change across large spatial (up to 240 km$^2$) and temporal (recent decades) scales. This study quantified reef loss in terms of seafloor height and reef volume reduction for five reefs across the Western Atlantic and Pacific through comparison of high-resolution Lidar-derived digital elevation models (DEMs) from the late 1990s to 2000s with bathymetric records dating as far back as the 1930s. A major advantage of this approach is that it allowed quantitative assessments of the cumulative result of all processes impacting reef height (see Box 1). Estimates for loss of reef surface elevation are significantly greater than estimates derived from carbonate budget conversions for the same region (Figure 4). However, the approach did include various

non-coral dominated habitat types (e.g. sand, seagrass, or sediment) not typically covered by census studies. The early baseline data have also been of suboptimal resolution for this type of comparison. Nonetheless, the approach illustrates the potential for larger-scale assessments of reef surface elevation change. A crucial question with regards to such large-scale efforts, however, is whether the large spatial and temporal coverage supports robust statistical inference to match the resolution and precision accuracy that could otherwise be achieved by in-field approaches. It can reasonably be assumed that these methods will miss a degree of granularity in capturing specific short- to mid-term temporal processes at the millimetre to centimetre scale (Teague et al., 2022).

At present, the trade-off between spatial coverage and measurable resolution (Table 1) thus still constrains the use of Lidar as a standard tool for studying specific aspects of reef accretion dynamics (Hedley et al., 2016). A key challenge in assessing vertical change from repeat bathymetric surveys is ensuring that all inter-temporal measurements are tied to a consistent and reliable vertical datum. Referencing surveys to a common tidal datum is essential, as it provides a stable baseline relative to sea level, allowing meaningful comparisons of elevation change over time. A range of studies, however, highlight its potential through large-scale assessments of relevant metrics including reef-scape bathymetry, topographic complexity and rugosity or bottom roughness at metre to sub-metre resolution (e.g. Hedley et al., 2016; Asner et al., 2020; Dee et al., 2020; Goodman et al., 2020; Harris et al., 2023; Li and Asner, 2023). Notable studies have also integrated remote sensing techniques for spatial upscaling of reef carbonate production estimates (Andréfouët and Payri, 2001; Brock et al., 2006; Moses et al., 2009; Leon and Woodroffe, 2013; Hamylton et al., 2013, 2017; Doo et al., 2017; Hamylton and Mallela, 2019), which illustrates the potential for large-scale reef accretion estimation through carbonate conversion methodologies (Figure 3).

### High-resolution temporal reef surface monitoring utilising photogrammetry

Akin to large-scale LiDAR imaging, emerging underwater imaging tools are being increasingly incorporated within standardised monitoring efforts to assess a wide array of reef geomorphic parameters (Magel et al., 2019; Remmers et al., 2024). The increased quality of consumer-grade cameras, alongside simplified analytic workflows, has facilitated the wide-spread application of close-range structure-from-motion (SfM) photogrammetry. This visualisation technique uses a series of overlapping photographs to construct high-resolution photomosaics or digital surface models, such as dense point clouds, gridded digital elevation models, or meshes (Westoby et al., 2012). Whilst sacrificing some degree of spatial coverage compared with the techniques discussed in the previous section, the achievable resolution makes SfM one of the most promising emerging approaches for quantifying contemporary reef accretion (Remmers et al., 2024). For photogrammetry, in-water activity is essentially limited to the placement of reference markers and the collection of imagery data (Burns et al., 2015b). Image material can be collected reasonably fast and at medium spatial resolution (10s–100s of square metres) by SCUBA divers, snorkelers, towed systems (Hatcher et al., 2020), autonomous underwater vehicles (Friedman et al., 2012), and even non-submerged drones (Casella et al., 2022). User-friendly modelling software tools are now readily accessible for model generation (e.g. Agisoft Metashape Professional or open-source alternatives), cross-model comparison (CloudCompare) (Lange and Perry, 2020) and automated annotation (TagLab) (Pavoni et al., 2022). These

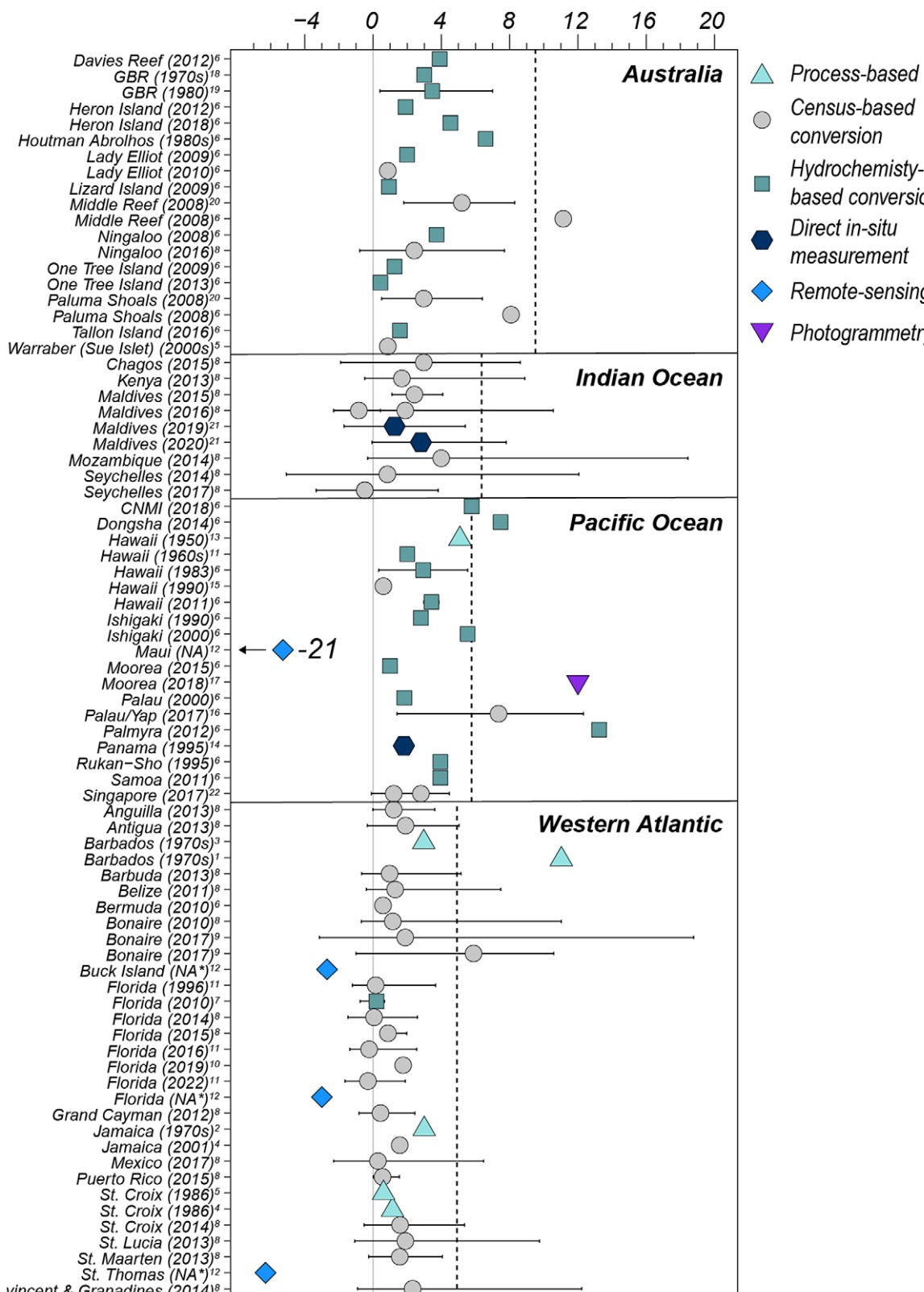

**Figure 4.** Published rates of modern vertical coral reef accretion (mm yr$^{-1}$) displayed by geographical region (panels) in relation to the Holocene average accretion rate (dotted line) (rates adapted from Hynes et al., 2024). Coloured symbols represent different methods used to estimate accretion rates, with error bars indicating standard deviations where available. Year of data collection given in brackets. No distinction is made for the reef zone or morphology. Site data and citations (superscript numbers) can be found in the Supplementary Table S1. *Data collected over several decades.

facilitate relatively intuitive image processing and data extraction pipelines.

3D – or more correctly referred to as 2.5D (e.g. Fukunaga and Burns, 2020) – reconstructions of entire reef communities or individual components are now enabling the extraction of fine-scales (cm-to-mm resolution) data on a range of relevant geomorphic and topographic features such as rugosity, surface roughness, mean height and reef-scape surface complexity (Figueira et al., 2015; Leon et al., 2015; Agudo-Adriani et al., 2016; Ferrari et al., 2016; Storlazzi et al., 2016; Magel et al., 2019; Bayley and Mogg, 2020; Hatcher et al., 2020; Aston et al., 2022; Zhong et al., 2023). The level of detail and scale that these models can provide outcompetes most traditional survey techniques in terms of resolution and time–cost-efficiency (Burns et al., 2015a; Curtis et al., 2023; Carneiro et al., 2024; Barrera-Falcón et al., 2025). The high repeatability of SfM and data permanence further implies significant potential for close monitoring of known and yet to be identified facets of structural change (Kopecky et al., 2023). At the colony level, SfM workflows designed to study the temporal evolution of structural metrics at submillimetre resolution are already well-developed. Rates of either growth or erosion can be quantified by aligning two models of the same colony taken at different time points (Ferrari et al., 2017; Royer et al., 2018; Olinger et al., 2019; Kodera et al., 2020; Lange and Perry, 2020; Ferrari et al., 2021; Million et al., 2021; Olinger et al., 2021; Lange et al., 2022; Lange et al., 2024). Automated algorithms then compute the distance between tie-points in both models (Lague et al., 2013) to calculate height changes in time across an entire relevant surface, rather than relying on a limited number of point measurements (e.g. CRAF or ReSET). However, expanding this technique for application at habitat or reef scales, while also retaining millimetre-scale precision, remains challenging (Remmers et al., 2024). This is because generating and aligning more spatially or structurally complex models still typically results in error margins that exceed the magnitude of elevation change over short (multi-year) time scales (Figueira et al., 2015; Hatcher et al., 2020; Nocerino et al., 2020).

Neyer et al. (2018) applied SfM to isolated areas of prominent changes in seafloor topography (~0.5 m) in large reef plots (>100 m$^2$) in Moorea, resulting from movement of large rubble fragments or changes in the distribution of sand. However, they struggled to accurately quantify less distinct growth of, for instance, living colonies. More recent efforts within the same Moorea region reviewed technical aspects of model creation for inter-temporal comparison but refrained from providing rates of growth (Nocerino et al., 2020). To our knowledge, only Rossi et al. (2020) have used SfM approaches to actually quantify change in mean reef height (in mm yr$^{-1}$), basing this on 25 m$^2$ plots in Moorea. They applied the Multiscale Model to a Model Cloud Comparison (M3C2) algorithm (Lague et al., 2013) to compute height change. They then manually verified these through the assessment of 2D stacked bathymetric profile sections to identify individual processes (e.g. colony accretion, erosion, branch fragmentation). However, recorded changes were predominantly driven by coral branch extension (or breakage), which is unlikely to directly translate into an equivalent rate of net long-term vertical reef accretion (Box 1).

Reliable inter-temporal repeatability in high spatial resolution models also presently relies heavily on a well-distributed underwater geodetic network of ground control points (GCPs) to scale and align two photogrammetric models (Skarlatos and Agrafiotis, 2018; Neyer et al., 2018; Hatcher et al., 2020; Rossi et al., 2021). While a few recent studies have demonstrated mm to cm precision in GCP alignment across short temporal scales (~1 yr), this has not yet yielded rates (in mm yr$^{-1}$) of reef height change (Hatcher et al., 2020; Nocerino et al., 2020; Zhong et al., 2023). This is often attributed to cross-model errors induced by slight variations in the camera equipment settings, distance to the object or movement or loss of GCPs (Nocerino et al., 2020; Rossi et al., 2020). Errors may also arise from artefacts in model generation and comparison, producing significant outliers when inter-temporal tie-points are mismatched. This may simply emerge due to minor alignment issues or because a coral branch broke off, leading to a high colony branch tip in model A being tied to a seafloor point in model B. The occurrence of such artefacts generally increases with topographic complexity (Figueira et al., 2015) or the abundance of moving objects such as algae, gorgonians, soft corals, or fish (Palma et al., 2018; Rossi et al., 2021). These issues can often be manually resolved, but this significantly increases post-modelling processing time. Careful adaptation of data collection to local environmental conditions may, at least partly, reduce such model errors.

## Concluding remarks and future perspectives

As ecological degradation continues to disrupt carbonate production, it will inevitably lead to an associated decline in vertical accretion rates. This will not only further limit reef-building and the many ecological functions that are tied to reef structural development, but also increasingly compromise the capacity of reefs to sustain reef islands and provide effective wave protection benefits. A central focus of this review has been on discussing recent methodological developments aimed at refining our ability to capture changes in the accretion potential of contemporary reef systems. What emerges from an overall assessment of these studies is that they collectively indicate that contemporary reef accretion rates are typically below long-term (Holocene averages) for the major coral regions (Figure 4). Throughout the Western Atlantic, this development can generally be attributed to widespread loss of coral cover. For many Indo-Pacific sites, however, this may in part be attributed to sea levels having remained relatively static for much of the mid- to late-Holocene, thus limiting shallow-water accommodation space for more recent vertical reef accretion (Smithers et al., 2006; Woodroffe and Webster, 2014). Whether modern rapid rates of SLR may reinvigorate vertical accretion in such locations remains uncertain, but the overall prognosis is that widespread ecological decline will likely lead to reduced accretion rates in most regions (Brown et al., 2011a; Chen et al., 2018).

Given this apparent global transition to low reef accretion rates, there is thus a critical need to develop robust frameworks for better constraining reef accretion potential. This will be important not only to better inform coastal risk models but also to guide appropriate restoration strategies. It is evident that historical records alone are no longer sufficient to reasonably project the accretion dynamics of future reefs. Primarily, this is because they lack information on the nonlinear and divergent responses of reef assemblages to anthropogenically induced disturbances, such as extreme heatwaves, mass coral disease outbreaks and the impacts of ocean acidification (Perry and Morgan, 2017; Toth et al., 2023a).

The few existing studies projecting future reef accretion rates under relevant climatic scenarios largely rely on information collected over recent annual to decadal scales and are typically based on carbonate budget conversions. These studies, including global (Cornwall et al., 2021), regional (Kennedy et al., 2013) and detailed site-specific assessments (Webb et al., 2023), suggest that accretion rates on many reefs will likely be insufficient to keep pace with

projected SLR under future climate change. Critical to refining these modelled projections further will be the integration of coral adaptive capacities and species-specific responses to warming and acidification. Addressing these gaps will require more nuanced parameter data collection tailored to the regional variability in environmental change and species responses (Cornwall et al., 2023; Webb et al., 2023).

Predicting the timescales over which diminished reef accretion may lead to significant geomorphic changes in coastal habitats is also highly complex and will vary geographically with the level of reef resilience. Ecological decline is already driving reef geomorphological changes in many locations by significantly reducing reef relief (Lewis, 2002; Alvarez-Filip et al., 2009). In areas where this reduction is severe, reefs may no longer provide significant wave-energy attenuation. This, in turn, could drive morphological changes in adjacent habitats such as sandy shores, lagoons and mangrove forests, over timescales ranging from years to decades. Noticeable geomorphic impacts on reef-driven island formation and maintenance may take longer to manifest, although this process will likely be exacerbated by SLR (Kench, 2024).

Advancing our predictive capabilities will depend on systematically addressing the limitations of current methods to quantify reef accretion capacities while also leveraging technical advances. Here, the potential benefits of integrating emerging imaging tools and automated processing pipelines in contemporary reef accretion monitoring are enormous (Hamylton, 2017; Parsons et al., 2018; Kopecky et al., 2023). As these techniques continue to mature, rapid advancements in the quality of tools used across all processing steps – from data collection to model extraction – are expected to resolve many of the technical challenges outlined above (e.g. Parsons et al., 2018; Wei et al., 2018; Kutser et al., 2020; Purkis and Chirayath, 2022; Caballero and Stumpf, 2023; Menna et al., 2024). However, caution is needed to ensure that critical information is not overlooked by removing the surveyor from the field. For instance, studies like those of Yates et al. (2017) are excellent for scaling up reef accretion assessments under contemporary global conditions, but it remains crucial to consider the individual small- and midscale processes that underpin the observed dynamics. Furthermore, some aspects that are crucial for capturing the detailed and context-specific processes that affect reef accretion cannot yet be effectively captured remotely and still rely on detailed in-field assessments. These include data from cryptic and mesophotic environments, environmental explanatory variables and the density of the studied substrates.

Overall, the best approaches for approximating modern and future vertical reef accretion potential likely involve a combination of in situ assessments and emerging techniques. Such integrated approaches will allow for the largest spatial coverage possible while providing the detail and context-specific information necessary to inform model projections. For example, high-resolution census-based studies calibrated with in situ measured substrate change rate data for the most common substrate types could be upscaled using large-spatial scale habitat maps derived from drone or satellite imagery. Such approaches would provide the best combination of habitat-specific data but at more meaningful spatial scales. Rate data from fossil records, although increasingly disconnected ecologically from many modern reef states, will remain important for providing historical rate contexts. Of specific relevance here are records from historically stressed or marginal reef sites, as these can offer insights into accretion responses to environmental challenges or for low-coral cover communities. As technologies and methods continue to advance, it remains crucial to collect data on all accretion parameters in a systematic and comprehensive manner. Adequately integrating estimates of modern reef accretion parameters into habitat restoration and coastal risk mitigation strategies will then benefit from leveraging the strengths of each discussed approach (Perry et al., 2018; Ferrari et al., 2021; Toth et al., 2022, 2023b).

**Open peer review.** To view the open peer review materials for this article, please visit http://doi.org/10.1017/cft.2025.10005.

**Supplementary material.** The supplementary material for this article can be found at http://doi.org/10.1017/cft.2025.10005.

**Data availability statement.** All presented data are sourced from published articles, and an overview of the literature used, along with the extracted accretion rates, is provided in the Supplementary Material.

**Acknowledgements.** We thank the reviewers for their constructive feedback on the manuscript. For the purpose of open access, the author has applied a Creative Commons Attribution (CC BY) licence to any Author Accepted Manuscript version arising from this submission. All illustrations and figures in this article were prepared using Adobe Illustrator.

**Author contribution.** Conceptualisation: D.d.B., C.T.P., A.E.W.; Literature review and research: D.d.B. Writing and drafting: D.d.B., C.T.P.; Critical review and editing: C.T.P., A.E.W.

**Financial support.** This research was funded through a Leverhulme Trust Research Grant (RPG-2021-295) to C.T.P.

**Competing interests.** The authors declare none.

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
