## [Reviewer Report]

This is a thorough and comprehensive review on an important topic. I think it deserves publication with a few minor changes (listed below).

From looking at the author contribution statement, I anticipate that this is a literature review written by a PhD student (de Bakker) - it looks like the majority of the review and writing has been done by that student, therefore I suggest it may be more accurate to move forward with this as a single authored paper from the student.

Specific comments:

Abstract: The utility of studying reef accretion is framed in terms of ongoing benefits of reefs for coastal protection and ecosystem services, what about reef islands? These are homes for many coastal communities and accretion is fundamental to their existence.

Intro para 1: There is also the idea that as upper reef surfaces become inundated throughout the tidal cycle, they may be colonised by corals (thereby increasing accretion).

Intro para 2: What about shifts between different carbonate producers, i.e. from corals to Halimeda? It may be worth describing other components that are critical to ongoing overall accretion.

Terminology Box:

Why are corals primary pridcuers and other calcifiers secondary? This varies gepgraphically. There are sites (e.g. Raine Island on the GBR) where forams are the dominant producers.

Carbonate budgets section: The hydrochemical work of Donald Kinsey was formative for developing early methods of estimating production, and most of his papers seem to have been overlooked.

In-situ methods:

RSET frames have ten pins for measuring elevation, not a single pin as stated. This method also requires around 5 years of measurements. Reference is made to a photographic record being challenging, but this is not part of the method (further explanation required)

Section 4.2.1 header references optical methods, but this section discusses Lidar, which is not an optical method. This section would also benefit from discussing the fundamental problem of establishing an accurate vertical datum relative to tide against which two bathymetric records can be compared within a single (and reliable) frame of reference to establish vertical changes.

Javier Leon’s structure from motion work at Heron Island has been very influential for this methodology and deserves elaboration.

---

## [Reviewer Report]

Abstract:

The utility of studying reef accretion is framed in terms of ongoing benefits of reefs for coastal protection and ecosystem services, what about reef islands? These are homes for many coastal communities and accretion is fundamental to their existence.

Intro para 1: There is also the idea that as upper reef surfaces become inundated throughout the tidal cycle, they may be colonised by corals (thereby increasing accretion).

Intro para 2: What about shifts between different carbonate producers, i.e. from corals to Halimeda? It may be worth describing other components that are critical to ongoing overall accretion.

Terminology Box:

Why are corals primary pridcuers and other calcifiers secondary? This varies gepgraphically. There are sites (e.g. Raine Island on the GBR) where forams are the dominant producers.

Carbonate budgets section: The hydrochemical work of Donald Kinsey was formative for developing early methods of estimating production, and most of his papers seem to have been overlooked.

In-situ methods:

RSET frames have ten pins for measuring elevation, not a single pin as stated. This method also requires around 5 years of measurements. Reference is made to a photographic record being challenging, but this is not part of the method (further explanation required)

Section 4.2.1 header references optical methods, but this section discusses Lidar, which is not an optical method. This section would also benefit from discussing the fundamental problem of establishing an accurate vertical datum relative to tide against which two bathymetric records can be compared within a single (and reliable) frame of reference to establish vertical changes.

Javier Leon’s structure from motion work at Heron Island has been very influential for this methodology and deserves elaboration.

---

## [Editor Report]

Dear Didier,

Thank you for revising your article based on the comments supplied. I enjoyed reading the updated version and I am of the opinion that it is now sufficiently improved and can be accepted for publication in Coastal Futures. Congratulations!

However, I do have a further request relating to the figures. Several of the Figures are very busy with text (Figures 2, 3 and 4). There is too much information here for a reader to take in without being overwhelmed. It would be good for the text on these to be reduced and simplified before they can be embedded within the main body of the manuscript text. Please revise the figures in the final phase of production.

Thank you for taking the time to put this review paper together for Coastal Futures.

Best wishes,

Sarah